In silico detection of dysregulated genes and molecular pathways in Alzheimer’s disease as basis for food restoring approach

Petrignani Ilaria 1
Pasquo Alessandra 1
Bei Roberto 2
Di Nardo Paolo 2 3
http://orcid.org/0000-0002-0142-6690 Carotenuto Felicia 2 3
Pappagallo Noemi 4
Fraternale Daniele 4
http://orcid.org/0000-0003-4549-1475 Albertini Maria Cristina 4
Teodori Laura 1 teodori@med.uniroma2.it
1 Diagnostics and Metrology Laboratory (NUC-TECFIS-DIM), Research Center ENEA , Frascati, Rome , Italy
2 Department of Clinical Sciences and Translational Medicine, University of Roma “Tor Vergata” , Rome , Italy
3 Interdepartmental Research Centre for Regenerative Medicine (CIMER), University of Roma “Tor Vergata” , Rome , Italy
4 Department of Biomolecular Sciences, University of Urbino , Urbino , Italy
Menini Stefano
Electronic publication date: 2025 Apr 7
Publication date: 2025
Volume: 13
Electronic Location ID: e19100
Received 2024 Nov 7; Accepted 2025 Feb 12
Copyright: © 2025 Petrignani et al.
Copyright year: 2025
Copyright holder: Petrignani et al.
License: This is an open access article distributed under the terms of the Creative Commons Attribution License, which permits unrestricted use, distribution, reproduction and adaptation in any medium and for any purpose provided that it is properly attributed. For attribution, the original author(s), title, publication source (PeerJ) and either DOI or URL of the article must be cited.
License URL: https://creativecommons.org/licenses/by/4.0/

Keywords: Network biology, System science, Nutrition, MicroRNA, Age-associated disease

Funding: University of Urbino Carlo Bo 446 23.10.2020 European Union—Next Generation EU—PNRR MUR ECS_00000041-VITALITY—CUP J13C22000430001 This research was supported by the University of Urbino Carlo Bo (DISB_ALBERTINI_PROG_SIC_ALIMENTARE_2021-DISB D.R. n.446 23.10.2020) and by the European Union—Next Generation EU—PNRR MUR project ECS_00000041- VITALITY—CUP J13C22000430001. There was no additional external funding received for this study. The funders had no role in study design, data collection and analysis, decision to publish, or preparation of the manuscript.

==============================
Forty-eight million people worldwide suffer from dementia, often associated with the growth of the elderly population. There are also concerns about the younger population, where increasing acute and chronic abuse of alcohol and neurotoxic substances may contribute to brain damage and the early onset of dementia. Alzheimer’s disease (AD) accounts for 60% of dementia cases and most therapies used so far have been unsuccessful. Genetic, epigenetic and vascular factors contribute to the pathogenesis of AD. Among the epigenetic mechanisms, modulation of microRNA (miRs) plays an important role. To detect genes and pathways involved in AD, we performed an original bioinformatic analysis of published Alzheimer’s dysregulated miRs using MIcroRNA ENrichment TURned NETwork (MIENTURNET) followed by Reactome tools. The interrogation of these platforms allowed us to discover common putative genes (by MIENTURNET) targeted by the dysregulated miRs and the pathways in which the set of altered genes are involved (by Reactome tool). Our in silico analysis showed that the β-catenin phosphorylation cascade and Netrin-1 signalling, resulted as the most significant. Lastly, based on the assumption that food bioactive compounds (BC) modulate miRs, which in turn modulate dysregulated genes and pathways associated with AD, a literature search demonstrated that some BC are indeed able to modulate dysregulated pathways and genes. Curcumin, osthole, puerarin, xanthoceraside, sulforaphane, salvianolic acid A, resveratrol and andrographolide lead to upregulation of the Wnt/β-catenin pathway. Choline, methionine, folate and vitamin B6/B12 modulate the upregulation of the Netrin-1 pathway. In conclusion, our in silico analysis of miRs identified dysregulated genes and their associated pathways, paving interesting and new insights for diagnosis and for potential therapeutic interventions.

Introduction

The incidence of age-related diseases, including dementia, is increasing dramatically. Dementia affects 48 million individuals worldwide, with its incidence doubling approximately every five years between the ages of 65 and 90 (Alzheimer’s Association, 2025). In addition, serious concerns are also arising regarding younger population, where increasing acute and chronic abuse of alcohol and other neurotoxic substances is causing serious consequences and irreversible damage to the nervous system. Indeed, neuropathological and imaging studies suggest that alcohol abuse may lead to structural and functional brain damage, which might be associated with some forms of dementia (Ridley, Draper & Withall, 2013).

Alzheimer’s disease (AD) accounts for 60–70% dementia cases (Cummings et al., 2016) and it is considered as one of the major global health challenges: currently, 47 million individuals suffer from AD a figure projected to rise to the huge number of 131 million by 2050 (Prince et al., 2013). Many resources and hopes have been invested in AD research, but despite the extensive effort, no effective treatments are available to halt the progression, only some therapies exist to alleviate the symptoms. The lack of success in understanding the molecular mechanisms of AD has led several major pharmaceutical companies to discontinue research in Alzheimer’s and Parkinson’s pharmacology. Consequently, AD remains one of the most pressing challenges for neurobiology in the third millennium. In this challenging framework, it is of paramount importance to explore other strategies of intervention based on and substantiated by a greater understanding of the molecular mechanisms underlying AD onset and progression also using the most updated resources such as bioinformatic platforms, high throughput analyses and big data repositories. AD has multifactorial etiopathogenesis that includes sedentary lifestyle, diet, epigenetic factors, involvement of vascular and other systems (Kisler et al., 2017; Solis, Hascup & Hascup, 2020; Jara-Medina et al., 2024; Breijyeh & Karaman, 2020). The “two-hit hypothesis” provides a framework for understanding the role of vascular and other systems in AD. According to this hypothesis, the first “hit” involves vascular dysfunction, such as reduced cerebral blood flow and damage to the blood-brain barrier, which disrupts brain homeostasis and creates a permissive environment for neurodegeneration. The second “hit” is characterized by the accumulation of amyloid-β and tau proteins, leading to the formation of amyloid plaques and neurofibrillary tangles. These processes interact synergistically to accelerate disease progression and highlight the multifactorial nature of AD (Zhu et al., 2007).

Genetic factors, such as mutations in genes that regulate the metabolism of the amyloid precursor protein (the precursor of the β-amyloid peptide, the main component of amyloid plaques), and environmental factors have also been implicated in the pathogenesis of AD (Przedborski, Vila & Jackson-Lewis, 2003). The latter induces epigenetic modifications in our organism, leading to a change in gene expression without altering the nucleotide sequence. Epigenetics involves mechanisms that determine stable, heritable, but reversible changes in gene expression without altering the original DNA sequence. DNA methylation, histone modifications and microRNA (miRs) are the main epigenetic mechanisms involved in the pathophysiology of AD (Mastroeni et al., 2011; Paniri, Hosseini & Akhavan-Niaki, 2023). MiRs represent a promising key for many diseases diagnosis and therapy, including AD (Femminella, Ferrara & Rengo, 2015; Kaur et al., 2023). MiRs are small, non-coding, endogenous nucleotide sequences that negatively regulate the expression of their target genes in a post-transcriptional manner, inducing degradation or inhibition of target mRNA translation. Scientific evidence shows that some of these miRs are altered in patients with AD (Reddy et al., 2017).

In this work, the functions of miRs were investigated using bioinformatics approaches. Indeed, the advent of “systems biology” and “omics” technologies, the emergence of high-throughput data techniques and the existence of big data repositories allow the interrogation of many cellular features and the determination of the molecular network interacting with each other. It is thus possible to study biological systems through the acquisition of an extremely large number of molecular data and measurements, and to understand and model the properties of the multiplicity of networks that control cell behavior. This approach can “view” the scheme and the rules that govern the system to anticipate how the other players, connected to each other, are modified by the perturbation. It also represents a challenge for a branch of contemporary biology, network biology, which represents a promising field to understand cell structure and function.

Furthermore, with the continuous improvement of computational and bioinformatics tools, it is now possible to uncover the molecular mechanisms responsible for the pathogenesis of various diseases, such as signaling pathways, biological processes, genes, miRs involved, which are considered potential biomarkers. Many bioinformatics tools are now available to manage the flow of data. Most applications are accessible via an online interface. The MIcroRNA ENrichment TURned NETwork (MIENTURNET), Reactome and Search Tool for Interactions of Chemicals (STITCH) tools were used in this work and allowed us to predict the putative target genes of the modulated miRs in AD, the pathways in which these genes are involved and the intermediate molecules that interact with the target genes involved in the pathways of interest.

Recent evidence suggests that some nutrients and food bioactive compounds (BC), due to their ability to modify the expression/concentration of specific miRs, can protect the organism against certain diseases in which the alteration of their target genes is a fundamental part of the pathogenesis. Indeed, diet is an environmental factor that appears to be strongly related to AD through modification of epigenetic pathways (Athanasopoulos, Karagiannis & Tsolaki, 2016; Martínez-Iglesias et al., 2023). If altered miRs expression contributes to the development of AD, there is a possibility that “correcting” dysregulated miRs may be able to reverse the pathological process. Based on this assumption, nutrition plays an important role through modulation of miRs. Recent works showed the ability of natural bioactive compounds to regulate miRs expression (Vrânceanu et al., 2022). Bioactive compounds are a heterogeneous group of substances commonly consumed with daily diet, able to positively influence health, contributing to the prevention of different diseases, even if they cannot be considered nutrients in the strict sense. They are widely distributed in fruit, vegetables, legumes, whole grains, nuts, seeds, mushrooms, herbs and spices and in plant-based beverages such as wine and tea (Barbieri et al., 2017). The analysis of the properties of bioactive compounds opened the broad field of “functional food”.

Through an original bioinformatic analysis (in silico) of the available data, this article aims to: (1) investigate dysregulated genes and molecular pathways in AD, and (2) map natural and food-related compounds that can modulate the expression of altered miRs associated with AD (and their dysregulated pathways).

This work aims to answer new biological and medical relevant questions in AD. We used bioinformatic tools to discover new modulated gene targets and pathways related to miRs selectively up-regulated and down-regulated in AD.

Materials & methods

A literature search was performed using the PubMed free online database for the collection of AD dysregulated miRs. Scientific article with experimental evidence of modulation of miRs in AD (up- or down- regulated) have been collected using the following keywords: “microRNA and neurodegenerative disease” or “microRNA and Alzheimer’s disease”. The search resulted in a list of 3,166 articles, only articles published in English were included in this search (n = 3,128). The studies considered for inclusion were both human in vivo and clinical, and in vitro studies from 2008 to 2021 (n = 2,189). In vitro studies are regarded as solid tissues, biological fluids, and cell culture, all from human origin. The studies included the modulation of miRNAs (n = 46) (Fig. 1). Studies that used animal or cellular models not relevant to the research context, such as those involving species or tissues not comparable to humans, were excluded.

Figure 1 Flowchart of study selection for miRs modulation in Alzheimer’s disease.

The flowchart illustrates the workflow used to select studies providing experimental evidence of modulated miRs in AD. It represents the systematic process for identifying, screening and including relevant articles, ensuring that only studies that met defined criteria for experimental validation of miRs modulation in AD were considered.

In order to study the interactions between pathways or miRs and BC present in food, herbs etc., we searched the literature for BC capable of modulating the miRs obtained from MIENTURNET and Reactome pathways analyses. Thus, we collected published data focusing on the detection of BC with therapeutic value to promote the restoration of dysregulated pathways or miRs.

In silico (bioinformatics) analysis

Bioinformatic analysis was conducted using MIENTURNETand Reactome tools.

MIENTURNET was used to analyze the interactions between miRs and genes involved in the etiology of AD (http://userver.bio.uniroma1.it/apps/mienturnet/) (Licursi et al., 2019). This tool is an easy-to-use web tool that receives an input list of miRs and uses “miRNA-target interactions” computationally predicted and experimentally validated downloaded from TargetScan and miRTarBase, respectively.

An input list containing the ID of each miRs, obtained from miRBase (https://www.mirbase.org/), was entered in MIENTURNET that allows the resulting genes (with their statistical significance) to be visualized as miR-target network interactions. The tool processes the input file and provides in the ‘IDs found’ drop-down menu the total number of miRs recognized from the miRBase database along with a table that includes the name of the miRs, the accession ID in miRBase of the miRs, as well as hyperlinks to miRBase and the sequence of the mature miRs. Unknown miRs can be found by clicking on the ‘IDs not found’ drop-down menu. By clicking the miRNA-target Enrichment box, MIENTURNET performs a statistical analysis for over-representation of miRNA-target interactions included in the list of input miRs. This analysis evaluates whether interactions are significantly frequent in the input list. On the miR-target Enrichment page, it is possible to select the database to refer for the enrichment analysis of miR-target interactions (TargetScan and/or miRTarBase). We considered TargetScan appearing as the most up-to-date tool for sequence-based miRNA-target predictions. By checking the TargetScan box, computationally predicted miRNA-target interactions was considered. TargetScan predicts the biological targets of miRs by searching for conserved sites, matching with the “seed region” of each miRNA (Agarwal et al., 2015; McGeary et al., 2019).

Subsequently, we used Reactome database to identify the pathways of related genes. Reactome is a free, open-source, open-data, curated and peer-reviewed pathways database, which includes bioinformatic tools for the visualization, interpretation and analysis of pathways to support basic research (Gillespie et al., 2022). The central component of Reactome is the pathway, since it visually represents how nucleic acids, proteins, complexes and other small molecules are involved in biological pathways (apoptosis, energy metabolism, innate and adaptative immune response etc.). We used the “Analysis Tools” feature of Reactome and inserted the list of gene resulting from the previous MIENTURNET analysis.

Reactome identifies for each input list of genes entered, the pathways in which the genes are involved and selects the top 25 most significant pathways (Pathways Analysis Report).

After the detection of statistically significant genes and pathways we searched for BC able to modulate AD dysregulated pathways and genes including the dysregulated miRs obtained from the literature search (which were our input data). Interrogating the STITCH database, we were able to find intermediate molecules that interacted with the most significant important target genes previously found through MIENTURNET analysis. STITCH collects and integrates data from different sources to explore and predict interactions between compounds and proteins. It uses experimental data, databases and literature to find chemicals that are associated with other chemicals and proteins. In the present study, the genes of interest were entered into the STITCH database and human species were set up to screen the target proteins that interact with them (Szklarczyk et al., 2016). In Fig. 2 we summarized the sequence of our bioinformatic experimental procedure.

Figure 2 The workflow outlines the sequential steps of this study.

PubMed search was performed to identify dysregulated miRs in AD. MIENTURNET query: the list of upregulated and downregulated miRs in AD were inserted into MIENTURNET (input data) to identify target genes (output). MIENTURNET output was exploited as input to the reactome tool to discover common putative pathways. PubMed search to identify BC (Bioactive Compounds) able to modulate the identified pathways and genes involved in AD through MIENTURNET and Reactome. STITCH database interrogation for intermediates of the genes involved in the pathways obtained through Reactome interrogation.

Results

MIENTURNET analysis

MiRs that were up- or down-regulated in AD as retrieved from literature, were analyzed using the MIENTURNET platform. The input data we entered were the pool of experimental and clinical published data on AD, in which the levels of miRs were assessed in human samples and cellular models (Table S1). To perform the analysis, TargetScan platform (a feature of MIENTURNET) was chosen as the reference database for “miR-target interactions”. For down-regulated miRs, out of the 140 miRs inserted, 77 were recognized by the platform and analyzed. The remaining 66 miRs were not recognized and reported in the “IDs not found” panel of MIENTURNET. For up-regulated miRs, 152 were recognized and analyzed out of 155 miRs inserted (three of them were not present in the database). The input lists used for the analysis in MIENTURNET are provided in Table S2 (input-list miRs DOWN) and Table S3 (input-list miRs UP). We obtained the genes target of the dysregulated miRs uploaded on the platform. The results are illustrated in Tables S4 and S5, where the most significant altered genes obtained from MIENTURNET analysis selected on the basis of their p-value are shown. The MIENTURNET platform, in addition to providing the target genes of the inserted miRs, also provides the list of all (so far known) involved miRs (set of miRs that have the same gene as target). The results are presented in Table S6, where the genes targeted by down-regulated miRs and their target gene together with their p-value are indicated.

In Table S7 the target genes of up-regulated miRs, their interaction and their p-value are reported.

Reactome analysis

We used all the genes of Table S4 and S5 obtained from MIENTURNET analysis as input for Reactome analysis.

The results of the Reactome analysis, showing the pathway analysis reports for genes targeted by the down-regulated and up-regulated miRs, respectively, are shown in Supplemental Data S8 and S9. The results yielded β-catenin phosphorylation cascade pathway (Supplemental file S8) and Netrin-1 signaling (Supplemental file S9) as the most significant ones (Fig. 3; p-value 0.003).

Figure 3 Reactome analysis.

(A) The main pathways identified by inputting the target genes of miRs down-regulated in AD. (B) The main pathways of upregulated miRs in AD (right panel) are indicated. A cut-off of 0.005 was chosen for the analysis.

As illustrated in Fig. 3, β-catenin phosphorylation cascade pathway (Fig. 3A) and Netrin-1 (Fig. 3B) were the most significant putative dysregulated pathways resulted from the Reactome analysis.

Furthermore, from the Reactome Pathway Analysis Report, the β-catenin phosphorylation cascade pathway showed that FRAT2 and PPP2R5E are target genes of down-regulated miRs in AD (Supplemental file S8). As a result of the MIENTURNET enrichment analysis (Table S6), FRAT2 seems to be a target gene of the following down-regulated miRs: hsa-miR-221-3p, hsa-miR-22-3p, hsa-miR-23a-3p, hsa-23b-3p, hsa-miR-26a-5p, hsa-miR-26b-5p, hsa-miR-29b-3p. Furthermore, PPP2R5E seems to be targeted by hsa-miR-132-3p, hsa-miR-212-3p, hsa-miR-133b, hsa-miR-148a-3p, hsa-miR-181c-5p, hsa-miR-200a-3p, hsa-miR-221-3p, hsa-miR-23a-3p, hsa-miR-23b-3p, hsa-miR-301a-3p, hsa-miR-342-3p and hsa-miR502-3p.

The pathway mediated by Netrin-1 signaling showed that the NEO1 gene is modulated by the up-regulated miRs in AD (Supplemental file S9).

The enrichment analysis we performed by MIENTURNET, shows that NEO1 is a target gene of the following up-regulated miRs in AD (Table S7): hsa-miR-125a-5p, hsa-miR-125b-5p, hsa-miR-128-3p, hsa-miR-128-5p, hsa-miR-216a-3p, hsa-miR-216a-5p, hsa-miR-216b-5p, hsa-miR-27b-3p, hsa-miR-374a-5p, hsa-miR-429, hsa-miR-92a-3p, hsa-miR-92b-3p, hsa-miR-9-5p.

In the Supplemental files S8 and S9 are also indicated all the additional pathways associated with down-regulated and up-regulated miRs respectively. For down-regulated miRs, for example, SUMOylation of DNA damage response and repair proteins (involving the POM121C and STAG1 genes) and Regulation of PTEN mRNA translation (involving the PTEN gene), with p-values of 0.006 and 0.007 respectively (Supplemental file S8) were found. For up-regulated miRs, the pathways identified were Formation of Editosome (involving the A1CF gene) and mRNA Editing: C to U conversion (also involving the A1CF gene), with p-values of 0.01 and 0.013, respectively (Supplemental file S9). However, these pathways are not reported here since their p-values exceed our set threshold (p < 0.005) but will be briefly addressed in the discussion section.

Bioactive compounds

To provide clinical support for our new findings reported we searched for the published literature for natural BC that may differentially modulate the pathways we obtained from bioinformatics analysis (i.e., β-catenin cascade phosphorylation and Netrin-1) and the dysregulated miRs in AD. STITCH analysis performed by inserting genes involved in the two pathways of interest, enabled the identification of intermediates (i.e., molecular mediators within the pathways). These intermediates include Glycogen Synthase Kinase-3 beta (GSK3β), Dishevelled proteins (Dvl), and the Netrin receptor DCC (DCC), which interact with the FRAT2 gene (GSK3β, Dvl) and the NEO1 gene (DCC) The network related to gene target FRAT2 and gene target of NEO1 are reported in Supplemental files S10 and S11.

In Figs. 4 and 5 we reported a schematic representation of food, BC and nutrients, found to be able to restore our pathways of interest (i.e., β-catenin and Netrin-1). Curcumin, osthole, puerarin, resveratrol, ginkgolide B, salvianolic acid A, andrographolide, xanthoceraside and sulforaphane are involved in neuroprotective role through activation of Wnt-β-catenin pathway, down-regulated in AD. The Wnt/β-catenin pathway is a broad signalling network within which the β-catenin phosphorylation cascade acts as a key regulatory mechanism, modulating the stability and activity of β-catenin. Some nutrients such as coline, methionine, B6/B12 vitamins and folate explain their potential neuroprotective effect by maintaining the level of Netrin-1.

Figure 4 Neuroprotective effect of bioactive compounds.

In the figure we schematically reported the natural compounds (curcumin, osthole, puerarin, resveratrol, ginkgolide B, salvianolic acid A, andrographolide, xanthoceraside, sulforaphane) that have the potential to enhance the down-regulated Wnt/β-catenin pathway in AD, resulting thus in a neuroprotective effect.

Figure 5 Role of nutrients in Netrin-1 pathway.

In the figure we reported a schematic representation of the role of nutrients in modulating the Netrin-1 pathway. As shown, an adequate intake of choline, methionine, vitamins B6/B12 and folate maintains the levels of Netrin-1 pathway expression that ensure neuroprotective effect.

In Fig. 6 we reported the BC able to modulate the expression level of miRs found altered in AD and involved in the pathways identified. We found that curcumin can up-regulate miR-22-3p, down-regulated in AD, and down-regulate miR-125b up-regulated in AD. Berberine induces up-regulation of miR-132a-3p down-regulated in AD.

Figure 6 Role of bioactive compounds on miRNAs modulation.

Schematic representation of bioactive compounds capable of modulating the expression levels of miRs found altered in AD which are involved in the identified pathways. Curcumin is able to up-regulate miR-22-3p, which is down-regulated in AD, and down-regulate miR-125b, which is up-regulated in AD. Berberine induces the up-regulation of miR-132a-3p, which is down-regulated in AD models.

Discussion

In this study, we conducted an in silico analysis, (i) through MIENTURNET interrogation of published data from different sources and (ii) using the results obtained from the MIENTURNET analysis as input for Reactome interrogation. This allowed us to identify (i) the target genes of AD dysregulated miRs and ultimately (ii) the pathways in which these genes are involved. We also successfully obtained insights into food BC and their impact on the dysregulated pathways obtained by Reactome. Indeed, as also demonstrated by others miRs may be modulated by BCs and nutrients (Vrânceanu et al., 2022). This modulation might, in turn, modulate dysregulated genes associated with AD, potentially restoring their expression to a healthy phenotype.

Defining interventions based on miRs, that simultaneously restore various altered pathways in AD, represent up to now one of the most promising prospects for AD treatment. The increasing evidence emphasizing the importance of miRs dysregulation in AD has led us to investigate the biological function of miRs to identify altered pathways associated with them.

Our in silico results highlight that the β-catenin phosphorylation cascade and Netrin-1 signaling pathways are the most significantly dysregulated in AD. Previous observations as well (Ju et al., 2022; Ramakrishna et al., 2023) also found Netrin-1 (Ju et al., 2022) and WNT-β Catenin dysregulation in AD, although they found decreased levels of WNT-β Catenin. Our bioinformatic analysis identified the β-catenin phosphorylation cascade as modulated and associated with downregulated miRs. Our analysis does not allow us to establish whether this pathway is up- or down-regulated, although in association with down-regulated miRs, the occurrence of other additional regulatory mechanisms, or the interplay of other miRs targeting the same gene, could inhibit the pathway despite the downregulation (as we find in our study) of the miRs that normally target it. Factors such as oxidative stress, neuroinflammation or epigenetic changes, all common in AD pathology, could contribute to the overall downregulation of β-catenin signaling, overriding the expected effects of miRNA downregulation. Additionally, recent findings indicate that, in certain cases, miRs may also upregulate gene expression (Orang, Safaralizadeh & Kazemzadeh-Bavili, 2014). This may further complicate the picture of the molecular networks involved in cell signaling and highlights the need for studies such as ours.

In the central nervous system, the Wnt/β-catenin pathway is essential in signal transduction, regulating numerous cellular processes like neuronal survival, neurogenesis, regulation of synaptic plasticity and integrity of the blood-brain barrier (Aghaizu, Jin & Whiting, 2020). Some evidence of the involvement of Wnt signaling pathway in AD, has been proposed (Jia, Piña-Crespo & Li, 2019). Caricasole et al. (2004) and Rosi et al. (2010) show that the canonical Wnt antagonist Dickkopf-1 (Dkk1) is up regulated in brain of AD patients and mouse models. Dkk1 inhibits canonical Wnt signaling by interacting with LRP5/6 Wnt co-receptors, thus impairing the binding of Wnt proteins to both Frizzled and LRP5/6. Inhibition of Wnt signaling by Dkk1 leads to increased Gsk3ß activity and reduced cytoplasmic β-catenin levels, both features which are observed in the brains of AD patients. The increase in Gsk3ß kinase activity, observed in the AD brain, contributes to the hyperphosphorylation of Tau protein, distinctive pathological trait of the disease (Salcedo-Tello, Ortiz-Matamoros & Arias, 2011). Overexpression of Gsk3ß in the brain leads to neurodegeneration and learning deficit. The increased activity of Gsk3ß promotes the degradation of β-catenin, leading to a reduction in the canonical pathway (Mateo et al., 2006).

Netrin-1 is an endogenous secreted laminin-related protein identified as a bifunctional neuronal guidance molecule, through its interactions with canonical receptors. It influences axonal growth and plays a key role in axon arborization, dendritic growth and synapse formation interacting with its receptors such as deleted in DCC and Uncordinated-5 (UNC5) (Xia et al., 2022).

Netrin-1 acts as a negative regulator of Amyloid-β (Aβ) (the main component of the amyloid plaques associated with Alzheimer) through its interaction with the APP (Lourenço et al., 2009). The reduction in Netrin-1 production in the brain of transgenic mice (considered as AD model), was associated with the increase in Aβ concentration. Aβ-induced neurotoxicity has been accepted as a hallmark component in the pathogenesis of AD (Borel et al., 2017; Rama et al., 2012). Decrease in Netrin-1 was also correlated to a Th17/Tregs (T helper 17/regulatory T cells) balance disorder in a rat model of Aβ-induced AD (Sun et al., 2019). Th17 cells are proinflammatory, while Tregs play an essential role in maintaining immunological homeostasis and regulating autoimmunity. In serum and cerebrospinal fluid of AD mouse model, the reduced level of Netrin-1 appears to be related with an increase in Th17 cells and reduction of Tregs. This imbalance leads to inflammatory process, a key component in AD pathogenesis (Kiraly, Foss & Giordano, 2023).

Besides Netrin and β-catenin, other pathways related to AD were also identified by our bioinformatic analysis but not considered here due to a less significant p-value.

The Reactome analysis of genes targeted by down-regulated miRs found: SUMOylation of DNA damage response and repair proteins and Regulation of PTEN mRNA translation.

SUMOylation is a post-translational modification that involves attaching SUMO proteins to lysine residues in proteins, regulating processes such as DNA repair, transcription, and stress response. In the context of neurodegenerative diseases, dysregulation of SUMOylation is linked to AD pathogenesis (Mandel & Agarwal, 2022). Studies in transgenic mouse models have shown changes in SUMO protein expression, particularly SUMO1, in cortical and hippocampal tissues, which are associated with learning and memory deficits (Krumova et al., 2011).

SUMOylation appears to modulate the formation of amyloid-beta (Aβ) plaques by modifying APP (Zhang & Sarge, 2008).

Regarding the pathway “Regulation of PTEN mRNA translation”, different studies evidence involvement of PTEN deregulation in neurodegenerative disease and AD (Ferrarelli, 2016). The study of Griffin et al. (2005) suggested that in AD there is an activation of the Akt/PKB pathway with increased phosphorylation of Akt substrates. At the same time, PTEN (a tumor suppressor and negative regulator of Akt) undergoes loss of function or altered distribution. This PTEN dysfunction leads to over-activation of the Akt pathway, which disrupts normal cellular processes such as neuronal survival, autophagy and protein degradation. These changes contribute to the accumulation of toxic proteins such as amyloid-beta and tau, which play an important role in the progression of AD pathology.

The Reactome analysis of up-regulated miRS found: Formation of editosome and mRNA editing: C to U conversion.

Editosome is a protein complex responsible for RNA editing, which alters RNA sequences after transcription. RNA editing, the process that alters individual bases of RNA, may contribute to AD pathogenesis due to its roles in neuronal development and immune regulation and could represent an important post-transcriptional regulatory program in AD pathogenesis.

Regarding the pathway, mRNA editing: C to U conversion, this is a specific type of RNA editing where a cytosine (C) is converted to uridine (U) in mRNA. The most common type of RNA editing is the deamination of adenosine (A) bases to inosine (I) by the Adenosine Deaminase RNA Specific (ADAR) family of enzymes, and the second is the deamination of cytidine (C) to uridine (U) by the activation induced cytidine deaminase (AID)/apolipoprotein B editing complex (APOBEC, is an enzyme in mammals that plays a specific role in RNA editing) cytidine deaminases (Polson et al., 1991; Bass, 2002). This modification can change the encoded protein. In AD abnormalities in this type of RNA editing may result in defective proteins involved in neuron protection, inflammation, or protein aggregation (like Aβ or tau). These faulty proteins could contribute to the progression of the disease, including memory loss and cognitive decline.

The A1CF gene-target of up-regulated miRs, is found in both pathways (i.e., editosome and mRNA editing). If the A1CF gene, which encodes the APOBEC1 is down-regulated, this can have significant consequences for the formation of the editosome and the C-to-U RNA editing process. When A1CF is down-regulated, the APOBEC1 enzyme may not be able to perform its C-to-U editing efficiently or accurately, leading to a reduction or loss of RNA editing at certain target sites. RNA editing, specifically C-to-U conversion, modifies protein sequences by changing the codons in mRNA. If this editing is disrupted due to a lack of A1CF, some proteins that depend on these changes after transcription may be produced in their original, non-functional forms. As a result, these proteins may become defective and unable to carry out their normal roles in the cell. In neurons, this could affect proteins that are essential for key processes such as communication between synapses, handling cellular stress, or degrading other proteins. All these processes and consequences, potentially contributing to the progression of Alzheimer’s disease by impairing neuronal function and increasing cellular stress and degeneration (Karagianni et al., 2022; Lerner, Papavasiliou & Pecori, 2018).

Considering the growing body of scientific evidence on natural products for treating various pathologies and the role of nutrition in maintaining health, we proposed an approach for modulating altered pathways (i.e., Wnt/β-catenin and Netrin-1) in AD. Through a search on PubMed, we found that many BC have been identified as potential modulators of the Wnt/ β-catenin and Netrin-1 pathway as reported in the results. Some of these BC (such as curcumin, resveratrol and sulforaphane) may be directly assumed through regular daily diet, from plant used for nutritional purpose Curcuma longa (turmeric), Brassicaceae (broccoli and cabbage) and Vitis vinifera (wine, grapes). On the other hand, medicinal plants such as Cnidium monnieri, Pueraria lobata, Xanthoceras sorbifolium, Salvia miltiorrhiza (danshen) and Andrographis paniculata are used primarily to treat specific medical conditions. Their compounds, such as osthole, puerarin, xanthoceraside, salvianolic acid and andrographolide, have pharmacological properties that make them effective in therapeutic contexts rather than nutritional ones. This distinction is crucial because bioactives from medicinal plants are often associated with controlled use, unlike compounds found in nutritional plants, which are safe for regular consumption.

Some molecular insights related to the BC found to be able to modulate the dysregulated pathways in AD are reported in Supplemental file S12.

Conclusions

Our in silico analysis allowed us to detect significant molecular dysregulated genes and pathways involved in Alzheimer’s disease (AD). Additionally, the identification of gene-nutrient crosstalk interactions using the STITCH tool opens promising new avenues for future research. Further exploration of these interactions may deepen our understanding of nutrient-gene dynamics in AD and lead to novel therapeutic strategies.

Our in silico research lays the foundation for the combined use of “data science”, appropriate repositories, databases, and bioinformatics algorithms in the therapeutic management of AD. The results and conclusions presented are both interesting and potentially valuable for further comprehensive investigations aimed at fully elucidating the effects of BC in the context of neurological diseases.

Supplemental Information

Supplemental Information 1 AD upregulated and downregulated miRs.

The modulated miRs (up-regulated or down-regulated) in AD based on a search of studies retrieved from the PubMed online database. The data presented in the table include results from human in vivo or clinical studies and in vitro investigations. MiRs identified as potential biomarkers or involved in the pathogenesis of AD are included in the table with their corresponding regulation and references to the original studies.

Supplemental Information 2 Input-list miRs down-regulated in AD, used for the analysis in Mienturnet tool.

Supplemental Information 3 Input-list miRs up-regulated used for the analysis in Mienturnet tool.

Supplemental Information 4 Mienturnet analysis, target genes of up-regulated miRs.

The target genes of up-regulated miRs in AD. The most significantly altered genes, selected based on their p-value, were further analyzed using the Reactome online tool.

Supplemental Information 5 Mienturnet analysis, Target genes of downregulated miRs.

The target genes of down-regulated miRNAs in AD. The genes showing the most significant alterations, identified by their p-values, were subjected to further analysis using the Reactome online tool.

Supplemental Information 6 Mienturnet analysis.

Target genes for each miRs down-regulated from the input list.

Supplemental Information 7 Mienturnet analysis.

Target genes for each miRs up-regulated from the input list.

Supplemental Information 8 Pathways analysis report conducted by Reactome.

The following table shows the 25 most relevant pathways sorted by p-value, related to the target gene of down-regulated miRs in AD.

Supplemental Information 9 Pathways analysis report conducted by Reactome.

The 25 most relevant pathways sorted by p-value, related to the target gene of up-regulated miRs in AD.

Supplemental Information 10 Network related to gene-target FRAT2 (red ball) and other molecules from STITCH database.

The analysis was carried out considering Homo sapiens as the organism and a high confidence level of 0.700. Stronger associations are represented by thicker lines. Protein-protein interactions are shown in grey, chemical-protein interactions in green and interactions between chemicals in red. Node size indicates structural knowledge: small nodes correspond to proteins with unknown 3D structure, while large nodes represent proteins with known or predicted 3D structures (http://stitch.embl.de/cgi/network.pl?taskId=kYKGpVE071V4).

Supplemental Information 11 Network related to gene-target NEO1 (red ball) and associated genes obtained from STITCH database.

The analysis was carried out considering Homo sapiens as the organism and a high confidence level of 0.700. Stronger associations are represented by thicker lines. Protein-protein interactions are shown in grey, chemical-protein interactions in green and interactions between chemicals in red. Node size indicates structural knowledge: small nodes correspond to proteins with unknown 3D structure, while large nodes represent proteins with known or predicted 3D structures (http://stitch.embl.de/cgi/network.pl?taskId=qdIkJad1IB3L).

Supplemental Information 12 Insights related to bioactive compounds and pathways.

Molecular insights related to the BC found to be able to modulate the dysregulated pathways in AD.

Additional Information and Declarations

Competing Interests

Maria Cristina Albertini is an Academic Editor for PeerJ.

Author Contributions

Ilaria Petrignani conceived and designed the experiments, performed the experiments, analyzed the data, prepared figures and/or tables, authored or reviewed drafts of the article, and approved the final draft.

Alessandra Pasquo performed the experiments, analyzed the data, authored or reviewed drafts of the article, and approved the final draft.

Roberto Bei analyzed the data, authored or reviewed drafts of the article, and approved the final draft.

Paolo Di Nardo analyzed the data, authored or reviewed drafts of the article, and approved the final draft.

Felicia Carotenuto analyzed the data, authored or reviewed drafts of the article, and approved the final draft.

Noemi Pappagallo performed the experiments, prepared figures and/or tables, authored or reviewed drafts of the article, and approved the final draft.

Daniele Fraternale analyzed the data, authored or reviewed drafts of the article, and approved the final draft.

Maria Cristina Albertini conceived and designed the experiments, analyzed the data, authored or reviewed drafts of the article, and approved the final draft.

Laura Teodori conceived and designed the experiments, performed the experiments, analyzed the data, authored or reviewed drafts of the article, and approved the final draft.

Data Availability

The following information was supplied regarding data availability:

The bioinformatics data are available in the Supplemental Files.

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
