# Peer review of "In silico detection of dysregulated genes and molecular pathways in Alzheimer’s disease as basis for food restoring approach"

_PeerJ, doi:10.7717/peerj.19100_

## Round 0.1 · original submission · Major Revisions

Dear Dr. Teodori

Your manuscript entitled "In silico detection of dysregulated genes and molecular pathways in Alzheimer’s disease as basis for food restoring approach", which you submitted to PeerJ, has been reviewed by the editor and 2 experts in the field.

The reviewers are generally favorable but have raised significant concerns that must be addressed before the manuscript can be considered further. I would be willing to reconsider if you wish to undertake substantial revisions and resubmit.

If you decide to resubmit the revised version, please summarize all the improvements made in the new version and give answers to all critical points raised in the reviewers’ report in an accompanying letter. Copy and paste each and every reviewer's comment above your response. Please consider these points carefully, as the revised manuscript will undergo a second round of review by the same reviewers.

I hope you will be prepared to make the necessary amendments and submit a revised manuscript with a statement of how you responded to the reviewers’ comments.

Yours sincerely,

Stefano Menini

·

Basic reporting

Use correct punctuation in the whole manuscript. For example, rephrase line ll 183; For our study, we used.
Line ll 250; No data on gene expression regulation by BC 251 were found. Check all noun-verb matchings.

Paragraph writing is not pursued over whole manuscript. Please avoid short sentences.

Experimental design

Please add the script of advanced research in PubMed using determined Boolean logics to better be reproducing your research and data curation of third party. Your script is not written right, and it has typos. For example, Alzheimer’s disease not Alzheimer disease.
((miroRNA) AND (neurodegenerative disease)) OR ((micoRNA) AND (Alzheimer's disease)) hits 3048 results from 2008. Please specify the time range of study.
Please add a PRISMA diagram https://www.prisma-statement.org/prisma-2020-flow-diagram regarding exclusion/inclusion criteria for selecting relevant papers that were used for data distillation.
Please pursue above mentioned workflow for curating microRNA and biocompounds.
Specify that, why you selected two pathways for bioactive compounds?
Specify the website addresses of MIENTURNET, TargetScan, and miRTarBase and suffixes of its input and output files to be fast reproducing by readers and reviewers.
Please report codes or manually conversion of microRNA list to ID using miRBase. Put all codes of project in GitHub and give its link in your manuscript draft for better reproducing your projects.

Specify the p-value that you considered in REACTOME and screening your top-list.

Validity of the findings

Discussion must be architected based on the novelties of this study and shortened as possible. Please avoid discussing as introducing facts and general data, focusing on your achievements in this study.

Additional comments

Make this manuscript as short as possible to increase readership. I spent 6 hours reviewing this draft, please avoid redundancy.

·

Basic reporting

• writing errors in the texts (see file).
• Figures are relevant. However, the authors do not include integrative pathways with sites modified with microRNAs. They used a Reactome platform, these tools give the possibility to include all signaling to induce AD.
• Intro & background require a definition of Alzheimer's disease and show all hypotheses involved in the development of diseases, including two hit hypotheses.
• Literature well referenced, but requires all pathways of MAPK and their connection with Beta-catenin/WNT signaling for a more clear connection with food. Please include the bioinformatic paper Jara-Medina K, Lillo L, Lagunas C, Cabello-Guzmán G, Valenzuela-Melgarejo FJ. Identification of Vascular Genes Differentially Expressed in the Brain of Patients with Alzheimer's Disease. Curr Vasc Pharmacol. 2024;22(6):404-416. doi: 10.2174/0115701611298073240612050741. PMID: 38910465.

Experimental design

Incorporates original primary and bioinformatic research aligned with the journal's scope.

Clearly defines a well-articulated and relevant research question.

Demonstrates a rigorous investigation, adhering to high technical and ethical standards.

Provides detailed and transparent methodological descriptions, enabling reproducibility using bioinformatic platforms.

Validity of the findings

• All underlying data have been provided, which can be reproduced. But require a new figure connecting molecular pathways of AD – point of microRNA – Food effects.

Additional comments

• The introduction requires improvement. Several manuscripts showed the development of Alzheimer's disease is a multifactorial disease that includes, the vascular system, epigenetic, and other systems. Please see:
Kisler K, Nelson AR, Montagne A, Zlokovic BV. Cerebral blood flow regulation and neurovascular dysfunction in Alzheimer’s disease. Nat Rev Neurosci. 2017 Jul;18(7):419–34.
Solis E, Hascup KN, Hascup ER. Alzheimer’s Disease: The Link Between Amyloid-β and Neurovascular Dysfunction. J Alzheimers Dis JAD. 2020;76(4):1179–98.
Jara-Medina K, Lillo L, Lagunas C, Cabello-Guzmán G, Valenzuela-Melgarejo FJ. Identification of Vascular Genes Differentially Expressed in the Brain of Patients with Alzheimer's Disease. Curr Vasc Pharmacol. 2024;22(6):404-416. doi: 10.2174/0115701611298073240612050741. PMID: 38910465.
The authors used other manuscripts can be used here for all potentials mechanism.

---

## Round 0.2 · Minor Revisions

Dear Dr. Teodori,

Thank you for your resubmission. Below, I have included the comments from Reviewer 2, emphasizing additional points that need your careful attention before we can further consider your submission. I strongly encourage you to address each point comprehensively.

Please copy and paste each reviewer's comment above your corresponding response when revising your manuscript. Additionally, please provide a complete version of the manuscript with tracked changes to facilitate the verification of the revisions made.

If you are willing to do this, I would not need to return the manuscript to the reviewers, but it could be accepted for publication.

I look forward to receiving your revision,

Sincerely yours,

Stefano Menini

·

Basic reporting

Improved

Experimental design

Improved

Validity of the findings

Improved

·

Basic reporting

• Writing errors corrected.
• Figures modification with integrative pathways with sites modified with microRNAs.
• Intro & background defined Alzheimer's disease and show all hypotheses involved in the development of diseases, including two hit hypotheses.
• Literature well referenced.
• Abstract requires a modification due to the causes of Alzheimer's (AD). AD has multifactorial etiopathogenesis that includes several risk factors. Drugs and alcohol are two of several factors.

Experimental design

• Original primary and bioinformatic research within the Scope of the journal.
• Modification are OK.

Validity of the findings

"no comment".

Additional comments

Thank you for the modifications made to the manuscript. The revised version meets the conditions requested in the initial review. However, it is necessary to modify a paragraph in the Abstract to ensure consistency with the Introduction. Specifically, the paragraph in the Abstract (Lines 23-27) should be revised to align with the content in the Introduction (Lines 50-55; Cummings et al., 2016) and Line 66, where it is stated that “AD has a multifactorial etiopathogenesis that includes epigenetic factors, involvement of vascular and other systems (Kisler et al., 2017; Solis et al., 2020; Jara-Medina et al., 2024). The ‘two-hit hypothesis’ provides a framework for ………”
Abstract, Line 23-27.


“However, concerns are arising regarding the younger population, where acute and chronic abuse of alcohol and neurotoxic substances is increasing, resulting in permanent nervous system damage and the onset of dementia. Alzheimer’s disease (AD) accounts for 60% of dementia cases and most therapies used so far have been unsuccessful”
This phrasing suggests that alcohol and drug use in young individuals are the primary causes of dementia and AD, which is not entirely accurate. While substance abuse is a contributing factor, AD is a multifactorial disease influenced by several other elements, including vascular alterations, sedentary lifestyle, diet, and epigenetic changes, as discussed in the Introduction. The current sentence could be misinterpreted as prioritizing two factors over others, rather than acknowledging the complexity of AD's pathogenesis.
To ensure a more accurate and balanced presentation of risk factors, we kindly request a revision of this section in the Abstract to reflect the multifactorial nature of AD, as supported by the literature (Breijyeh Z, Karaman R. Comprehensive Review on Alzheimer's Disease: Causes and Treatment. Molecules. 2020 Dec 8;25(24):5789. doi: 10.3390/molecules25245789. PMID: 33302541; PMCID: PMC7764106)

---

## Round 0.3 · accepted · Accept

Dear Dr. Teodori,

Thank you for submitting the revised version of your manuscript. After a thorough review of the changes by Reviewer 2 and myself, I am pleased to inform you that all the reviewers' comments have been adequately addressed. Therefore, your manuscript is ready for publication in PeerJ.

I thank all reviewers for their efforts in improving the manuscript and the authors' cooperation throughout the review process.

Sincerely yours,

Stefano Menini

·

Basic reporting

Improved the abstract and introduction.

Experimental design

no comment

Validity of the findings

no comment

Additional comments

The manuscript is improved. Thanks.